# Quality Control of Cell Lines Using DNA as Target

**José Antonio Carrillo-Ávila** [†], **Purificación Catalina** [†] **and Rocío Aguilar-Quesada** *

Andalusian Public Health System Biobank, Coordinating Node, 18016 Granada, Spain;
jantonio.carrillo@juntadeandalucia.es (J.A.C.-Á.); purificacion.catalina@juntadeandalucia.es (P.C.)
* Correspondence: rocio.aguilar.quesada@juntadeandalucia.es
† These authors contributed equally to this work.

**Abstract:** Cell lines are a widely used pre-clinical models for biomedical research. The accessibility and the relative simplicity of facilities necessary for the use of cell lines, along with the large number of potential applications, encourage many researchers to choose this model. However, the access to cell lines from a non-confident source or through the interlaboratory exchange results in uncontrollable cell lines of uncertain quality. Furthermore, the possibility of using cell lines as an endless resource through multiple passages can contribute to this uncontrolled scenario, the main consequence of which is the lack of reproducibility between the research results. Different initiatives have emerged to promote the best practices regarding the use of cell lines and minimize the effect on the scientific results reported, including comprehensive quality control in the frame of Good Cell Culture Practice (GCCP). Cell Banks, research infrastructures for the professional distribution of biological material of high and known quality and origin, are committed with these initiatives. Many of the quality controls used to test different attributes of cell lines are based on DNA. This review describes quality control protocols of cell lines whose target molecule is DNA, and details the scope or purpose and their corresponding functionality.

**Keywords:** cell lines; good cell culture practice; DNA-based quality control; Cell Banks

## 1. Introduction

Although guidelines for the use of cell lines in biomedical research including quality control have been published [1] and international standards related with the quality of cell lines have been defined and published [2–7], more efforts are necessary to promote the best practices in order to contribute to the reproducibility and credibility of experimental results.

Compliance with Good Cell Culture Practice (GCCP) is essential to have quality-controlled material independently of its nature or downstream application [6–9]. The aim of GCCP is the establishment of principles for the standardization and implementation of practices for cell and tissue culture according to Good Laboratory Practice (GLP), in relation with characterization and maintenance of essential characteristics, quality assurance, recording, reporting, safety, education and training, and ethics [8]. Thereby, GCCP was initially developed to define minimum standards in cell and tissue culture due to the limited applicability of GLP (originally addressed for only regulatory in vivo studies) to in vitro work and research environments, facilitating implementation of practices linked to the application of the principles of GLP and providing guidance to research laboratories, journals, and funding bodies. Accordingly, GCCP is periodically revised to incorporate new cell technologies and approaches [9].

The importance of Cell Banks' activity for quality and cell accessibility has been highlighted [9,10] in the frame of accredited and authorized facilities [11]. Cell Banks certify the quality and traceability of cell products processed and stored, for both the Master Cell Bank (MCB) and the Working Cell Bank (WCB) [4,12]. Quality controls of cell lines include the tools necessary to monitor their authentication, stability, functionality, and contamination absence by using different molecular biology techniques [1,4]. Several

levels and checkpoints of quality controls have been proposed and implemented in order to optimize costs or resources. Thus, some authors have classified quality control testing in "Critical Release" and "Informational Testing" [3], others have identified those quality controls useful in a specific framework [6], whereas others apply quality controls by levels or steps corresponding to the attributes tested along the workflow pipeline [13,14].

Specifically, this review focuses its attention on quality controls of cell lines based on DNA as a target, describing their corresponding functionality, and scope or purpose. Thus, well-established approaches of quality control based in DNA, some of them recognised as standards, are presented along this review.

Several techniques to analyse genomes at different resolution levels are available and can be used as a quality control of cell lines. These techniques can cover the full genome or be restricted to a small part, so that none can meet all the requirements of range, resolution, sensitivity, and low cost. In this review, different technologies including G-banded karyotyping, array comparative genomic hybridization (aCGH) and other microarray approaches such as single nucleotide polymorphism (SNP) arrays, or DNA methylation are revised. Methods to analyse small parts of the genome by means of the design of probes or primers (e.g., fluorescence in situ hybridization (FISH) and polymerase chain reaction (PCR)) allow to detect known recurrent abnormalities, to perform DNA fingerprinting or to measure telomere length. Additionally, microbiological control of cell lines can also be carried out due to a microbial DNA analysis.

Therefore, this review promotes quality controls through DNA analysis, with an up-to-date and unbiased assessment of the indications for the use of the different techniques, their applicability as quality control based on DNA, establishing recommendations for their implementation.

## 2. Conventional Cytogenetics: Karyotyping

The karyotype is defined as the accurate organization (matching and alignment) of the chromosomal content of any given cell line. In a karyotype, chromosomes are arranged and numbered by size, from the largest to the smallest one. The karyotype is the normal nomenclature used to describe the normal or abnormal, constitutional, or acquired chromosomal complement of an individual, tissue or cell line. The most common cell karyotype study method used for years is the conventional or classic cytogenetic, where the cells are cultured, metaphases are obtained, and the crossed chromosomes with an appropriate stain are studied. The G-banded karyotyping is the cytogenetic conventional assay commonly employed in both clinical and research settings. This technique, although it was first described many years ago, is still valid and informative [15]. Karyotyping evaluates the entire genome at a resolution of $\geq$5–10 Mb and can detect abnormal subpopulations as low as 14% mosaicism [16] assuming a standard analysis of twenty cells metaphases [17,18]. Although sub-microscopic variants beyond karyotype resolution can occur, karyotyping remains a versatile assay, especially when complemented with high-resolution testing [19].

The conventional cytogenetic study has been widely considered the gold standard for genetic testing since it has been the most available to evaluate the whole karyotype at the same time. Nevertheless, it is subject to limitations as only dividing cells can be assessed, and analyses are expensive because of the lack of automation in sample processing and the time needed to analyse each cell division or metaphase. Trained and expert personnel are necessary to perform a valuable analysis. Moreover, there is no useful result from some cell lines if these results are not analysable, there are no cell divisions, or just a few divisions can be analysed due to it is not always possible to obtain an adequate number of metaphases in the process or the quality of these metaphases does not permit a detailed study of the chromosomes. For these reasons, currently there are other state-of-the-art methods of molecular cytogenetics that can resolve the main problem of this conventional technique.

The karyotype analysis is supported by the International System for Human Cytogenomic Nomenclature (ISCN) [20]. This the central reference for the description of

karyotyping, FISH, and microarray results, and provides rules for describing cytogenetic and molecular cytogenetic findings in laboratory reports.

As quality control of cultured cell lines, the conventional cytogenetic techniques are used routinely for the determination of numerical chromosomal abnormalities or structural rearrangements, mainly translocations, to quickly identify chromosomal alterations.

## 3. Fluorescence In Situ Hybridization (FISH)

The introduction of molecular cytogenetic techniques, based on FISH, revolutionized the field of cytogenetics [21] by allowing the identification of complex and cryptic chromosomal alterations, without prior knowledge of chromosomal loci involved. FISH allows the study of chromosome exchanges and gene rearrangements, copy number alterations, amplification, and deletions at the single-cell level, in a complementary way to conventional cytogenetics.

FISH is based on fluorescently labelled DNA probes that hybridize to unique DNA sequences along the chromosomes, on either metaphase preparations or interphase cells. Probes and target DNAs are denatured using high-temperature incubation in a formamide or salt solution. The probe is applied in excess, so the kinetics ensure that the probe anneals to the target DNA. Probe detection is accomplished by ultraviolet-light excitement of a fluorochrome, such as fluorescein-5-thiocynate (FITC) or tetramethyl rhodamine isothiocyanate (TRITC), which is directly attached to the DNA probe, or by incubation of a hapten labelled probe with a fluorescent conjugate. The majority of probes used for laboratory purposes are commercially available [22].

There are wide applications of FISH, mainly in cancer research and molecular diagnosis, but also as quality control for DNA stability detection of cell lines. However, issues related with long experimental times, expensive reagents and the requirement of trained technicians must be addressed in order to increase FISH applications in cytogenetic analysis. In this sense, the protocol recently evolved towards on chip detection of chromosome alterations through microsystems for FISH analysis, reaching automation of the assay performance, reduction in probe volume, as well as reduction in assay time [23].

Another of the FISH technique limitations is that a small number of genes can be simultaneously quantified. Thus, to increase the throughput of this informative technique, a fluorescent barcode system for the unique labelling of dozens of genes and simultaneous hybridization, and an automated image analysis algorithm, were devised. The reliability of this multiplex approach has been demonstrated on normal human lymphocytes, metaphase spreads of transformed cell lines, and cultured circulating tumour cells. It also opens the door to the development of gene panels for more comprehensive analysis of copy number changes, including the study of heterogeneity, and of high-throughput clinical assays that can provide rapid quantification of gene copy numbers in samples with limited cellularity, such as circulating tumour cells [24].

On the other hand, although FISH was originally used for chromosome analysis, it is currently being replaced by aCGH or next generation sequencing (NGS). For the diagnosis of single gene defects, PCR has become highly developed and is being used. Moreover, SNP arrays for karyomapping also were recently introduced [25].

## 4. Copy Number Variations (CNVs) Detection

Copy Number Variations (CNVs), defined as genomic intervals that deviate from the normal diploid state (deletions or duplications), were collectively detected in an estimated 4.8 to 9.5% of the human genome [26]. This includes both population-specific and individual-specific variation. An average genome contains 3 to 7 rare variant CNVs. Between 5 and 10% of individuals have CNVs larger than 500 kb in size, and 1–2% of the population carry CNVs greater than 1 Mb in size [27]. Genome sequencing studies have shown that most bases that vary among genomes reside in CNVs of at least 1 kilobase (Kb). Population-based surveys have identified thousands of CNVs, most of which, due to lim-

ited resolution, are larger than 5 kb [28]. However, CNVs can range from sub-microscopic events to complete chromosomal aneuploidies.

The size of a CNV and its gene density are strongly anti-correlated with its frequency. Benign CNVs are often small, intergenic, or encompass genes that can tolerate a change in copy number, whereas pathogenic CNVs (those larger than 250 kb) are significantly enriched for genes involved in development and genes with constrained evolutionary patterns of duplication and loss, or strongly associated with morbid consequences such as developmental disorders and cancer. Therefore, their functional impact has been demonstrated across the full range of biology, from cellular phenotypes, such as gene expression, to all classes of human disease with an underlying genetic basis: sporadic, Mendelian, complex and infectious [29]. Detecting CNVs within and between populations is essential to better understand the plasticity of genome and to elucidate its possible contribution to disease or phenotypic traits [28].

Professional guidelines have been developed for the interpretation and reporting of clinically relevant CNVs, and consider factors such as size, genomic content, comparison with internal and external databases for population frequency information, and whether the CNV is inherited or de novo [30].

Diverse technologies, including aCGH and SNP microarrays, and more recently, whole genome sequencing and whole exome sequencing, have enabled robust genome-wide unbiased detection of CNVs in affected individuals and in reportedly healthy. The aCGH obtains better results than the G-banded karyotype analysis through analysis of the entire genome at a much higher resolution (1 kb in size). It was first introduced to the clinical diagnostic field with arrays that contained bacterial artificial chromosome (BAC) clones corresponding to known clinically relevant microdeletion genomic intervals. Genome advancements led to replacement of BAC clones by oligonucleotide sequences in the aCGH technology or by SNP probes.

The aCGH and SNP arrays enable genome-wide detection of CNVs in a high resolution, and therefore microarray has been recognized as the prioritized test for the pathologies detection and genetic alterations resulting from modifications in the DNA sequence. The detection of CNVs through the SNP array analysis, enables the identification of stretches of homozygosity and thereby the possible identification of recessive disease genes, mosaic aneuploidy, or uniparental disomy (UPD) if contained within a single chromosome. The patient-parent trio analysis is used to screen for the presence of any form of UPD in the patient, and can determine the parental origin of a de novo copy number variation [31].

However, limitations of chromosomal microarrays, both aCGH and SNP arrays, include the inability to identify copy-neutral rearrangements such as balanced rearrangements, inversions, or Robertsonian translocations. Moreover, they do not provide information regarding location of a gained copy in the genome (insertions) nor of its orientation (directly repeated or inverted) [29]. Alternatively, evolution of aCGH technology is important, highlighting exonic SNP arrays developed to detect small intragenic copy number changes, as well as large DNA fragments for the region of heterozygosity [32].

On the other hand, sequencing of breakpoint junctions allows to elucidate the upstream mechanisms leading to genomic instability and resultant structural variation, whereas studies of the association between CNVs and specific diseases or susceptibility to morbid traits have enhanced understanding of the downstream effects [29].

Furthermore, a recent technology that may fit between aCGH and NGS is next-generation mapping of long (from 100 bp to mega-base pairs) fluorescently labelled DNA molecules imaged on nanochannel arrays, which can identify CNVs or structural abnormalities with high resolution and sensitivity [33].

Consistently, the outcome of a genotype analysis may be used as a quality control by ruling out cell lines cross-contamination. Moreover, it is a tool that can be used for the detection of singular fragments of the genome and that serves to identify a specific genome which would make it usable for the identification of cell lines with specific alterations. Accordingly, a simple approach compatible with routine use in the laboratory was devel-

oped for the accurate determination of cell line identity throughout the course of long-term research use by genotyping 34 SNPs [34]. Therefore, these higher resolution techniques are a very useful tool in the detection of incorrectly identified cell lines.

## 5. DNA Methylation

DNA methylation is a heritable epigenetic mark involved in gene regulation and cell differentiation characterized by direct chemical modification to the DNA at the cytosine residue of primarily cytosine guanine dinucleotides (CpGs) [35]. Global, gene-specific, and epigenome-wide methylation can be analysed by means of different methods, more or less laborious, or complex based on commercially available kits [36]. DNA methylation has a key role in the cell identity and differentiation [37]. Consequently, methylation profiles have been proposed for the classification into pluripotent and non-pluripotent cells and therefore as a quality biomarker of human induced pluripotent stem cells [38]. DNA methylation signatures have been related to the identity of cells, allowing the classification of human mesenchymal stromal cells and fibroblasts and the identification of tissue of origin, contributing to the quality control of these cultures [39]. DNA methylation changes are related with senescence and ageing resulting in epigenetic clocks [40]. Therefore, DNA methylation signature is identified to be used as quality control of replicative senescence [41]. Because DNA methylation levels and hypermethylation of certain genes occurs in cancer initiation and progression [37,42], the analysis of DNA methylation profiles might provide complementary information about tumorigenic potential of cells.

## 6. Telomeres Length Measurement

Human telomeres are nucleoprotein structures, located at the end of the chromosomes, with the main function of chromosomes protection and genome stability. Human cells telomeres are composed of repetitive hexanucleotide (TTAGGGn) combined with the protein complex shelterin, and assembled into macromolecular structures called telomere-loops [43]. Telomere length is different between different cell types, and a decrease in telomere length is clearly associated with senescence processes [44]. On the other hand, different mechanisms for telomere length maintenance are activated to promote a continuous cell division in oncogenic episodes [45]. In normal human somatic cells, telomeres range from 5 to 15 Kb in length [46].

A great variety of techniques are used for telomeres length determination. Some of those techniques are based on the use of cells, such as Quantitative-Fluorescence in situ Hybridization (Q-FISH) [47] and Flow-Fluorescence in situ Hybridization (Flow-FISH) [48]. However, the most sensitive and accurate techniques for telomeres length determination require previous DNA isolation. In that way, techniques as Quantitative PCR (qPCR) [45], Terminal Restriction Fragments (TRF) [49], and recently the Telomere Length Combing Assay (TCA) [47], are preceded by previous DNA isolation.

## 7. DNA Fingerprinting and Short Tandem Repeats (STRs) Profiling

A widespread use of DNA for the quality control of cell lines is the determination of the genetic fingerprint. This technique frequently used for human identification and forensic sciences can be also successfully applied to cell lines identification. Traditionally, different techniques were used for cell line identification [10,50], including non-based DNA techniques [51,52]. The first methods based in the DNA analysis was the chromosomal analysis by karyotyping [53], HLA haplotypes analysis [54] and sequencing of DNA barcode regions [55]. However, these technologies are in decline for these purposes, giving way to DNA analysis with higher resolution capacity.

The discovery of DNA hypervariable regions within genomes made it possible to identify each human cell line derived from a single donor. In this sense, hypervariable regions which consist of variable number tandem repeat (VNTR) units from minisatellite DNA, are capable of hybridizing to many loci distributed throughout the genome to produce a DNA "fingerprint" [56]. In spite of the intrinsic difficulties of DNA fingerprint,

subsequent advances in the technology gave rise to the use of microsatellite regions or Short Tandem Repeats (STRs) consisting of core sequences of 1–6 bp, repeated in a different number in each cell line. Because the polymorphism of STRs are hotspots for homologous recombination events, these markers display many variations in the number of the repeating units between loci in unrelated cell lines [57]. Currently, 56 different STRs loci with more than 800 variants have been identified throughout the genome and can be analyzed [58]. The STRs are inherited in a Mendelian fashion, that is, an individual receives one allele from each parent [12]. The number of repetitions that these alleles present is different for each chromosome from the paternal or maternal route, which allows obtaining numerical data for each of them. The succession of numerical data obtained from the analysis of the different STRs studied allows obtaining a unique numerical combination for each sample analyzed. In addition, these data are easy to share with other laboratories and with the scientific community in general. For those reasons, STRs profiling is recommended as an international reference standard for human cell lines identification [1,10]. For STRs analysis, primers with fluorescent tags are used for PCR amplification of the required STRs loci. Fragment size and number of repetitions are determined by capillary electrophoresis in a sequencer or fragment size analyzer with size standards and controls, and the analysis is performed with a specific STRs analysis software. Recently, it was determined that the combination of 13 loci is accepted as unequivocal authentication of the cell lines analyzed [59].

The STRs analysis is considered a gold standard and it is mandatory for laboratories to detect cross-contamination as a consequence of the accidental introduction of cells from another culture, resulting in a false cell line and therefore leading to the publication of false and irreproducible results of research and clinical trials. A clear example of contamination is Hep-2, laryngeal cancer cells used in 8497 articles, which were contaminated by HeLa cells (cervical cancer cell line) since 1967 [60]. A current international initiative, the International Cell Line Authentication Committee (ICLAC), has available on its website a registry of misidentified cell lines that have been published and used in different research studies; currently, 531 misidentified cell lines have been described [61]. In addition, this organization offers a series of online Databases and Search Tools for Cell Line STR Profiles, as well as educational and training modules to carry out an appropriate cell lines quality control implementation [62].

## 8. Microbiological Controls

Microbiological controls during the culture of cell lines guarantee the absence of any adventitious agents such as viruses, bacteria, and fungi [63], being DNA detection fundamental for most of the methodologies used.

Virus contamination is probably the least studied, although it has been detected for very different viruses such as Epstein–Barr virus (EBV) [64], xenotropic murine leukaemia virus-related virus (XMRV) [65], yellow fever virus [66], lymphocytic choriomeningitis virus, Hantaan virus or reovirus 3 [67], bovine viral diarrhoea virus (BVDV) [68], among others. Frequently, BVDV infections proceed from commercial contaminated foetal bovine serum lots used for cells culture, which entails serious drawbacks in the pharmaceutical industry [69].

Fungal contamination is also quite common in cell cultures. Eradication of fungi is especially difficult as effective antifungal agents are often cytotoxic to the cell line and fungal spores are much more likely to persist in cultures [70]. *Aspergillus* sp., *Penicillium* sp., *Sepedonium* sp. and *Botrytis* sp. have been detected in cultures [71].

Undoubtedly, the most frequent contaminations are caused by bacterial infections. A study from a Cell Bank revealed that 39% of specimens were contaminated. The major contaminating agents were *mycoplasmas* (19%), followed by mixed infection (8%), fungi (8%), and other bacteria (4%). *Bacillus* sp., *Enterococcus* sp., and *Staphylococcus* sp. were the main bacterial agents among various species (except *mycoplasmas*) [71]. Bacteria such as as *Coxiella burnetii* were detected in another study [72].

Thus, *mycoplasma* contamination is the major problem in cell cultures. *Mycoplasmas* and the related *Acholeplasmas* (both referred as *mollicutes*) are the smallest self-replicating bacteria and the most prevalent microbial contaminant of cells. These microorganisms pass through standard 0.22 μm filters and they are not affected by commonly used antibiotics in cell media in such a way that it can grow extremely high titres without producing any turbidity in the supernatants, seriously affecting the experimental results of cell viability, gene expression, cell morphology, metabolism, and growing rate [73]. *Mycoplasma* contamination may affect both the scientific results of cell culture-based research and the quality of biological medicines manufactured by cell cultures in the biopharmaceutical industry for therapeutic use [74–76]. Because of the magnitude of this problem, a periodic *mycoplasma* detection test must be performed in every cell line manipulated in the laboratory. In fact, scientific journals require free *mycoplasma* cell lines before accepting manuscripts for publication [1].

Different studies show high contamination rates with *mycoplasma*. The German Cell Lines Bank, DSMZ, informed that 31% of leukaemia-lymphoma cell lines were contaminated [10] and a different study showed 29.3% contamination of cell cultures [77]. These disturbing data are due to absent or inadequate testing in many laboratories [10].

The World Health Organization (WHO) proposed to harmonize assays for *mycoplasma* DNA detection [73,78]. Although different methods are available for *mycoplasma* testing, the most extended and sensitive methods are Nucleic Acid Amplification Techniques (NAT assays) with their different variations: quantitative, semiquantitative or qualitative [1]. NAT assays allow results in 2–3 h by using real-time PCR, the specificity is really high, and they detect most of the *mollicutes* species. The lower manipulation in real-time PCR assays, linked to the fact that PCR amplified tubes are never opened in the real-time PCR method in the laboratory, reduces drastically the risk of contamination. It was noted that real-time PCR results are semi-quantitative, being indicative of the grade of *mycoplasma* contamination in the cell culture.

## 9. Conclusions

Unsurprisingly, there is not a unique tool of quality control that provides all the necessary information about cell lines, but a complementary group of DNA-based methods (and non-based in DNA, whose scope was not reviewed in this manuscript) are available for the qualification of cells. Each method will allow the characterization of specific aspects of cells such as identity, stability, or safety that will be more or less restrictive in function of the downstream application of cultures. The goal of this review is to support the cells users in the selection of the most appropriate DNA-based quality control of cell lines in the frame of a specific attribute to be characterized and their application field in order to contribute to the best practices (Table 1 and Figure 1).

**Table 1.** DNA-based quality controls of cell lines.

| Quality Control (QC) | DNA Target | Purpose or Attribute | Recommendation by the Application Field | References |
|---|---|---|---|---|
| G-banded karyotyping | Chromosomal metaphases analysis (alterations ≥5–10 Mb in size) | Stability, authentication, and species cross-contamination | Gold standard Mandatory for research and therapeutic use | [17,18,20,22] |
| FISH | Unique DNA sequences along the chromosomes | Stability | Complementary use if needed | [21–24] |
| aCGH, SNP arrays or NGS | CNVs | Stability and authentication | Highly recommended for clinical applications | [26–28,30–34] |
| DNA methylation | Methyl group to the C-5 position of the cytosine ring of DNA usually in a CpG dinucleotide | Identity and differentiation, senescence and tumorigenicity | Optional for research (if others QC for the same attributes are performed) Recommended for clinical applications | [38,39,41,42] |
| Telomeres length measurement | Nucleoprotein structures composed by repetitive hexanucleotide (TTAGGGn) combined with the protein complex shelterin | Tumorigenicity and senescence | Optional | [45–49] |
| STRs profiling | DNA hypervariable regions consisting of core sequences of 1–6 bp, repeated in a different number | Authentication | Gold standard Mandatory for research and therapeutic use | [1,10,59] |
| Microbiological tests | Microbial DNA detection | *Mycoplasma* contamination Virus and fungal contamination | Mandatory periodically for research and therapeutic use Mandatory in clinical applications | [1,63,70,71,73] |

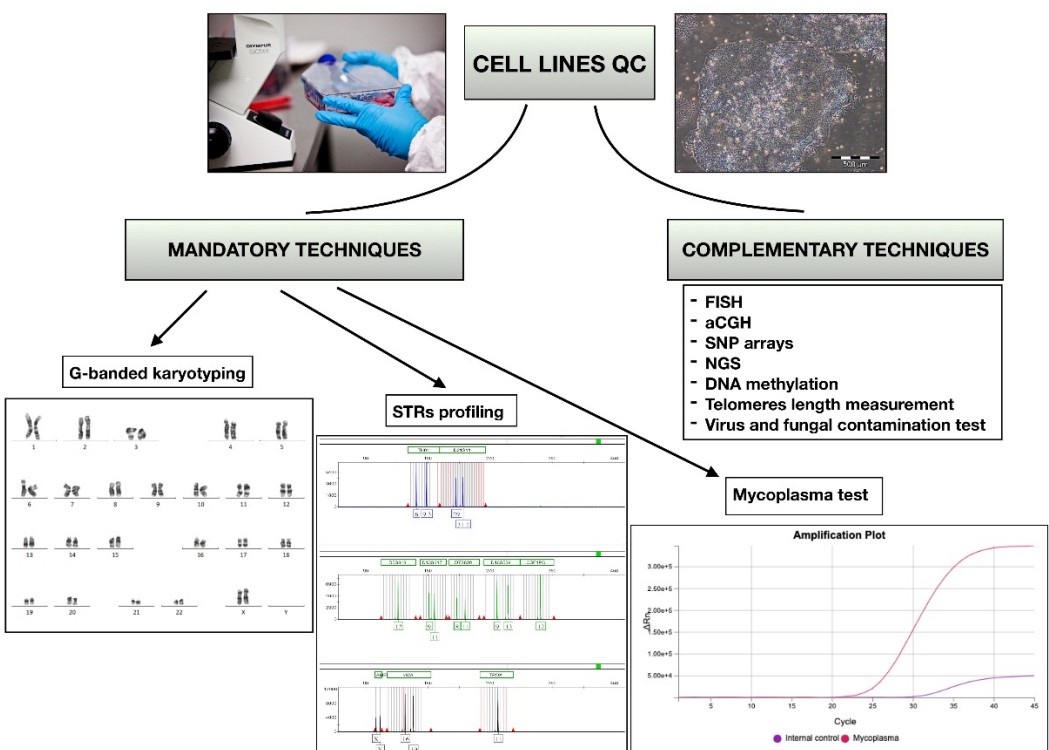

**Figure 1.** Recommendations of DNA-based quality controls of cell lines by the application field: mandatory for research and therapeutic use or complementary mainly addressed to clinical applications. Representative images from G-banded karyotyping, STRs profiling by fragment size analyzer after PCR amplification, and *Mycoplasma* test by real-time PCR are shown.

**Author Contributions:** Conceptualization, R.A.-Q.; writing—original draft preparation, J.A.C.-Á., P.C. and R.A.-Q.; writing—review and editing, J.A.C.-Á., P.C. and R.A.-Q. All authors have read and agreed to the published version of the manuscript.

**Funding:** This research was funded by PAIF from Consejería de Salud y Familias, Junta de Andalucía.

**Conflicts of Interest:** The authors declare no conflict of interest.

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
