# Peer review of "Quality Control of Cell Lines Using DNA as Target"

_2673-8856, doi:10.3390/dna2010004_

Round 1

Reviewer 1 Report

Please, re-check the manuscript carefully for mistakes or misprints.

Line (L) 11: insert “the” before interlaboratory,

L 15: insert “the” before “best”,

L 17, 22, 30: change “Practices” into Practice,

L 20:  change “controls” into “control protocols”,

L 22: change “controls” into “control”,

L 27: insert “the” before “best”,

Revise the sentence on L 39-41: “Quality controls are the tools necessary to monitor the authentication, stability, functionality, and contamination absence of cell lines.” to understand that the quality control of cell lines can include these several mentioned parameters for evaluation using different molecular biology techniques.

Revise using the abbreviations in the text to avoid repetition:

for example “SNP” for “single nucleotide polymorphism” because you have got several explanations, for example on L: 138, 167, 175 and 207 but you do not have at the beginning on line 55.

L: 166: delete the repetition of comparative genomic hybridization using only aCGH abbreviation,

L 315: change “filter” into filters,

L 316: change “mediums” into “media”,

L 349: insert “the” before “best”,

All names of bacterial and fungal species presented in paragraph 8. should be written using an italic type.

Reviewer 2 Report

The review manuscript is well written and, in my opinion, will be useful for both cell banks and laboratories, utilizing many cell cultures in a controlled environment.

Overall notes:

  • The article has no illustrations and only one table. I think that the information may be structured further to multiple tables with references inside the tables and, if possible, to some illustrations, depicting the typical analysis results.
  • The authors should describe directly the regulations implied on cell culture use – what area is covered by the GCCP, who should adhere to these rules, how does the GCCP match the GLP regulations?
  • If the analysis method is claimed to be mandatory, it will be better to state the rule, which requires this testing technique. Perhaps the needed illustration may be in the form of the flow chart – what should be done for characterization of various cell cultures?
